# Household Wealth Gradient in Low Birthweight in India: A Cross-Sectional Analysis

**DOI:** 10.3390/children10071271

**Published:** 2023-07-24

**Authors:** Bishwajit Ghose

**Affiliations:** Center for Social Capital and Environmental Research, Ottawa, ON K1M OZ2, Canada; bghose@cscer.ca

**Keywords:** birthweight, household wealth, maternal and child health, India

## Abstract

A low birthweight is a common complication that can result from numerous physiological, environmental, and socioeconomic factors, and can put babies at an increased risk for health issues such as breathing difficulties, developmental delays, and even death in severe cases. In this analysis, I aim to assess the differences in the burden of low birthweight based on household wealth status in India using data from the latest National Family Health Survey (NFHS 2019–21). The sample population includes 161,596 mother–child dyads. A low birthweight is defined as a weight that is <2500 g at birth. I used descriptive and multivariate regression analyses in R studio to analyse the data. The findings show that 16.86% of the babies had a low birthweight. At the state level, the percentage of low birthweights ranges from 3.85% in Nagaland to 21.81% in Punjab. The mean birthweights range from 2759.68 g in the poorest, 2808.01 g in the poorer, 2838.17 g in the middle, 2855.06 g in the richer, and 2871.30 g in the richest wealth quintile households. The regression analysis indicates that higher wealth index quintiles have progressively lower risks of low birthweight, with the association being stronger in the rural areas. Compared with the poorest wealth quintile households, the risk ratio of low birthweight was 0.90 times lower for the poorer households and 0.74 times lower for the richest households. These findings indicate that household wealth condition is an important predictor of low birthweight by which low-income households are disproportionately affected. As wealth inequality continues to rise in India, health policymakers must take the necessary measures to support the vulnerable populations in order to improve maternal and infant health outcomes.

## 1. Introduction

A low birthweight is defined as any newborn with a weight of less than 5.5 pounds (<2500 g) at birth, regardless of gestational age [1]. It is an important determinant of infant mortality, accounting for about 60% of all deaths in the first year among newborns [2]. As such, a low birthweight has many serious public health implications as it increases the risks of neonatal and infant death and disabilities, which translates to a higher burden on the healthcare system [3,4]. Infants who are born with a low birthweight are at a higher risk for developing serious medical conditions including neurological problems, respiratory issues, and infections [5]. Given the long-term health consequences, the economic implications of a low birthweight are also alarming. In addition to the increased cost of medical care that is associated with the illnesses, caregivers are often unable to spend as much time on income-earning activities to cope with the costs. In the context of low-income settings, such as the rural areas in India, local hospitals and clinics may not be equipped to provide the specialized care needs arising from a low birthweight. Due to its impact on developmental delays and cognitive impairments, a low birthweight can also hinder a child’s ability to grow and learn properly and can negatively impact their education and productivity during adulthood [6]. Additionally, a low birthweight was found to increase the risks of chronic diseases later in life including cardiovascular disease, diabetes, and hypertension, reducing life expectancy and quality of life [7,8,9]. These challenges warrant the need for preventive measures and interventions to reduce the prevalence of low birthweight, which can ultimately translate to healthier communities and lower healthcare costs.

Low birthweight is a multifaceted issue that can be caused by diverse physiological and environmental issues such as mothers’ exposure to smoking and drinking during pregnancy, poor nutrition, infections, and a lack of prenatal care, which is necessary for early detection of and addressing any pregnancy-related complications [10,11,12,13]. Apart from these issues, the socioeconomic conditions of families also play a major role in determining the health outcomes of mothers and children [14,15,16]. Socioeconomic aspects are often overlooked in medical care when it comes to health and well-being, although they are an integral part of the overall picture and can directly impact the effects of physical or environmental factors. For instance, households with lower incomes may struggle to afford healthy diets and access necessary medical care during pregnancy, leading to higher health risks for the mother and her fetus. Pregnant women who are malnourished may suffer from deficiencies (such as low iron levels) that can lead to anemia or other complications for both themselves and their unborn children [17]. Previous studies have shown that anemia is a major public health concern in India, with about 53.2% of non-pregnant women and 50.4% of pregnant women being affected by the condition [18]. Anemic pregnant women may experience fatigue, weakness, and shortness of breath, which can make it more challenging for them to carry out daily activities or engage in exercise.

Thus, pregnant women who are malnourished can face a multitude of health risks, including the possibility of developing dietary deficiencies. These deficiencies can have severe consequences for both the mother and her unborn child due to poor fetal development. In addition to affecting the maternal factors, being from a lower socioeconomic background can directly impact a child’s chances of having a healthy start in life. Research suggests that children from lower socioeconomic backgrounds are also more likely to face health disparities throughout their lives [19]. For instance, the prospective BELLA cohort study on the mental health of children and adolescents in Germany found that children with higher educated parents showed fewer mental health problems in a stressful life situation [19]. Maternal factors also play a crucial role in shaping a child’s well-being, and often, women from disadvantaged backgrounds have limited access to quality healthcare services during pregnancy. A lack of adequate prenatal care can lead to various complications for both the mother and the child, increasing the risk of preterm birth, low birthweight, and developmental delays [20]. Moreover, growing up in poverty means limited access to nutritious food options and proper healthcare. Children from lower socioeconomic backgrounds are more likely to live in food deserts or areas where healthy foods like fresh fruits and vegetables are not readily available. To address these socioeconomic issues, it is necessary to have a clear picture of the wealth-related differences in key health indicators such as birthweight. Therefore, in this study, I aim to analyse the prevalence of low birthweight at the national and state levels and explore the association between birthweight and household wealth status. The data were obtained from the National Family Health Survey, which is accessible to all registered users through the DHS program website [21].

## 2. Methods

### 2.1. Data Source

Data for this study were obtained from the fifth round of National Family Health Survey (NFHS 2019-21) [22]. The survey utilized a stratified multistage cluster sampling procedure to select a representative sample of households from across India. Firstly, the districts were stratified into urban and rural areas. The rural areas were further divided into smaller groups based on village population and percentage of scheduled castes/tribes. Some villages were selected from each rural group as primary sampling units. The urban areas were also divided into groups based on the percentage of scheduled castes/tribes. Secondly, 22 households were selected from each primary sampling unit using a systematic sampling method. A total of 30,456 primary sampling units were selected from 707 districts, and the survey was completed in 30,198 primary sampling units. In short, the sampling method involved stratifying the population, selecting primary sampling units from each stratum, listing households in each selected unit, and randomly selecting households from these lists. The survey was conducted in two phases between June 2019 and April 2021 by 17 field agencies, and a total of 724,115 women were interviewed, resulting in a response rate of 97%.

### 2.2. Description of the Study Variables

The outcome variable was birthweight, which was classified as normal and low birthweight. Newborns weighing less than 2500 g at the time of birth are considered low birthweight (LBW) babies, while those who weigh 2500 g or more at the time of birth are considered normal birthweight (NBW) babies. I used the following search strategy to find relevant articles on PubMed that guided the choice of the explanatory variables: (“Low Birth Weight”[Mesh] OR “Infant, Low Birth Weight”[Mesh] OR “Small for Gestational Age”[Mesh]) AND (“Socioeconomic Factors”[Mesh] OR “Social Class”[Mesh] OR “Wealth”[Mesh] OR “Poverty”[Mesh]) AND (“India”[Mesh]). Based on the review of the literature, the following variables were included in the analysis: residency (urban, rural) household wealth index quintile (poorest, poorer, middle richer, richest); ANC visits (inadequate/0–3, adequate/3+); wanting one last child (wanted one then, wanted one later, did not want more); age of women; years of education; total number of children ever born; caste (scheduled caste, scheduled tribe, other backward caste/OBC, none); religion (Hindu, Muslim, Christian, other).

The household wealth index serves as a composite measure of households’ financial situation and living standard, which is measured based on a household’s ownership of durable assets such as televisions and cars [23]. The index is calculated using principal component analysis, which assigns weights to each asset and housing characteristic based on their contribution to the household’s overall wealth. The resulting index is then divided into quintiles, with each household being assigned to one of five categories ranging from the poorest 20% to the richest 20% of the population.

## 3. Analysis

The analyses were performed using R and Stata version 17, using the svy command to account for the cluster survey design. The first step was to subset the dataset to select participants who provided data on the birthweight for the last child. The variable was then dichotomized and used to make cross tabulations to describe the percentage of LBW and NBW babies across the sociodemographic variables. The percentages of LBW babies by state and by wealth quintile were presented using a choropleth map and bar chart. Next, I ran three sets of a binary logistic regression analysis to calculate the risk ratios of the associations between birthweight and the explanatory variables (one for the full sample, and two more for the urban and rural samples) [24]. A *p*-value of <0.05 was considered statistically significant for all associations. Following the regression analyses, I calculated the variable importance plot to identify which attributes are the most important predictors of low birthweight. Lastly, I checked the variance inflation factor (VIF) [25] statistic to make sure there was no multicollinearity in the regression analyses.

## 4. Results

Table 1 presents the sample characteristics of the study population. The majority of the sample population (77.7%) was from rural areas. The poorest households accounted for the highest percentage of the sample (23.0%), followed by poorer households (22.7%) and middle-income households (20.2%). In terms of antenatal care (ANC) visits, 61.5% of the participants received adequate ANC visits. A small proportion of the participants reported wanting to have one last child later (3.7%) or not wanting any more children (3.4%). The mean age was 27.40 years, the mean highest year of education was 4.25 years, and the mean number of children ever born was 2.16. The majority of the sample belonged to other backward classes (OBCs) (40.7%), followed by scheduled castes (21.0%) and scheduled tribes (20.2%). The majority of the sample reported being Hindu (74.7%), followed by Muslims (14.0%), Christians (7.0%), and other religions (4.3%).

As shown in Figure 1, about 17% of all babies had a low birthweight in 2019–2021.

At the state level, the percentage of LBW ranges from 3.85% in Nagaland to 21.81% in Punjab, indicating that the incidence of LBW varies significantly across different states in India (Figure 2). Some other states with a relatively high percentage of LBW include Haryana (19.98%), Bihar (16.84%), and Assam (14.83%), and those with a low percentage include Manipur (6.14%) and Mizoram (4.54%).

Figure 3 shows that the percentage of LBW babies is higher among the poorest households in almost all states in India. For instance, the poorest households accounted for more than 50% of the LBW babies in Bihar and Jharkhand, and 46.67% in Nagaland. In contrast, the percentage of LBW babies is relatively lower among the richer population in most states. For instance, in Goa, about 51.20% of the richest households had NBW babies compared with only 2.65% in Bihar.

The regression analysis reveals several factors that are associated with the risk of low birthweight (Table 2). Rural residency is associated with a 7% lower risk compared to urban residency [RR = 0.93, 95%CI = 0.90,0.97]. Higher wealth index quintiles have progressively lower risks of low birthweight, from 0.90 [95%CI = 0.86,0.93] for the poorer quintile to 0.74 [95%CI = 0.70,0.78] for the richest quintile, indicating that wealth reduces the low birthweight risk, especially in rural areas. Attending adequate antenatal care visits is associated with an 8% lower risk of low birthweight [RR = 0.92, 95%CI = 0.90,0.95]. Births that are wanted later are associated with an 11% higher risk of low birthweight [RR = 1.11, 95%CI = 1.05,1.18], while mothers who do not want any more children are associated with a 15% higher risk [RR = 1.15, 95%CI = 1.07,1.24] of low birthweight. Every additional child born to the mother lowers the risk by 2% for a ratio of 0.98 [0.96,0.99]. Compared to scheduled castes, scheduled tribes have a 19% lower risk of low birthweight at [RR = 0.81, 95%CI = 0.78,0.85], OBCs have a 9% lower risk at [RR = 0.91, 95%CI = 0.88,0.94], and those with no caste have a 7% lower risk at [RR = 0.93, 95%CI = 0.89,0.96]. Muslim children have an 8% lower risk of low birthweight [RR = 0.92, 95%CI = 0.88,0.96], while Christian children’s risk is 43% lower at [RR = 0.57, 95%CI = 0.53,0.62] versus Hindu children.

As illustrated by Figure 4, the mother’s age and household wealth index were the most important predictors of birthweight in the sample population.

## 5. Discussion

The findings of this study highlight the significant role of wealth inequality in the prevalence of low birthweight in India. This study reveals that about one in five babies in India were of low birthweight in the 2019–2021 survey, and the prevalence of low birthweight varies significantly across the states. The percentage of low birthweight babies is higher among the poorest households in almost all states, indicating that poverty is a major determinant of low birthweight in the country. This finding highlights the need for targeted interventions to address the problem of low birthweight in different states based on their socioeconomic conditions. States with a high percentage of low birthweight like Punjab, Haryana, and Bihar need specific interventions to address the problem among the poorest households. Similarly, states with a low percentage of low birthweight, like Mizoram and Manipur, can serve as models for other states to learn from and replicate their successful strategies for reducing low birthweight incidences.

The wealth gap in the prevalence of low birthweight is probably linked to the various socioeconomic and environmental factors. The poorest households often face a range of challenges, such as limited access to nutritious food, inadequate healthcare facilities, poor sanitation, and environmental pollution, leading to the poor health status of the household members. According to previous studies, poor maternal health is one of the primary factors contributing to the higher incidence of low birthweight [26,27,28]. Women from low-income households often have limited access to quality healthcare facilities, including antenatal care, which can result in serious health conditions such as anemia, hypertension, and gestational diabetes [29]. These conditions can adversely affect fetal growth and development, leading to low birthweight. In line with previous studies, this study’s findings show that receiving adequate ANC visits has an inverse association with low birthweight [30,31]. Antenatal care visits provide opportunities for early detection and management of maternal health conditions that can adversely affect fetal growth and development. However, direct out-of-pocket costs needed to access quality healthcare services in some areas may be one of the main reasons why some expecting mothers cannot afford ANC visits, exposing themselves to greater risks during childbirth. Women from wealthier families generally enjoy better access to necessary healthcare, which plays a crucial role in improving birth outcomes. Access to routine prenatal care enables the management of health conditions and the early detection of potential complications before they become severe. This can significantly contribute to reducing the risk of low birthweight among newborns. Conversely, lower-income families in India face significant challenges in accessing adequate healthcare services due to financial constraints and a limited availability of healthcare facilities in their local areas. These limitations often result in delayed or inadequate prenatal care, which can have adverse effects on maternal and infant health. In short, this study’s findings emphasize the association between household wealth and birth outcomes, highlighting the significance of addressing inequities in the access to healthcare.

This study also highlights the importance of the timing and intention of pregnancies in the risk of low birthweight. Births that are wanted later are associated with a higher risk of low birthweight, while mothers who do not want any more children are also associated with a higher risk of low birthweight. This finding is consistent with an Ethiopian study that has shown the importance of family planning services in reducing the risk of low birthweight [32]. The study also confirms that the number of children that the mother birthed is an important determinant of low birthweight, with every additional child born to the mother lowering the risk of low birthweight. This finding suggests that parity can play a protective role in reducing the risk of low birthweight. In terms of caste and religion, the study found that scheduled tribes had a lower risk of low birthweight compared to scheduled castes, while OBCs and those with no caste also had a lower risk. This finding is consistent with previous studies that show the importance of caste and socioeconomic status in determining health outcomes in India. Muslim children were also found to have a lower risk of low birthweight compared to Hindus, while Christian children were found to have the lowest risk.

Income inequality is a growing social problem in India, especially when it comes to the health status of the population in low-income areas [33,34]. As the country continues to develop and place more emphasis on improving public health indicators, health policymakers should take immediate action to address this issue and ensure that all women have access to quality healthcare. The first step should be providing pregnant women from low-income households with adequate nutritional support and prenatal care, which can help to reduce maternal mortality rates by ensuring that pregnant women receive proper nutrition before and throughout their pregnancy, as well as regular checkups during doctor visits. Another important step is to increase access to affordable preventative care for all mothers, regardless of their financial situation. This could include providing basic vaccinations for infants and young children, as well as promoting healthy living practices such as proper hygiene and exercise habits for both mothers and their children.

This study is based on a large sample size and is nationally representative, which increases the generalizability of the findings to the entire population. The use of a regression analysis allowed for the identification of several factors that are associated with the risk of low birthweight, which can inform targeted interventions to improve maternal and child health outcomes. A downside of the study is the use of cross-sectional data, which prevents establishing any causality between the identified risk factors and low birthweight. Secondly, as the data are secondary, it was not possible to include information on some important factors that can affect birthweight, such as maternal nutritional status and other chronic health conditions.

## 6. Conclusions

In conclusion, the findings align well with the existing evidence that socioeconomic factors, access to care, maternal characteristics, and demographic backgrounds influence birthweight. A noticeable finding is that the household wealth situation was found to be the most important predictor among the non-modifiable risk factors of low birthweight. Addressing the financial and other socioeconomic determinants may help reduce health disparities in low birthweight. Overall, the findings suggest that efforts to reduce poverty and improve the social and economic status of households can play a crucial role in reducing the incidence of low birthweight in India.

## Figures and Tables

**Figure 1 children-10-01271-f001:**
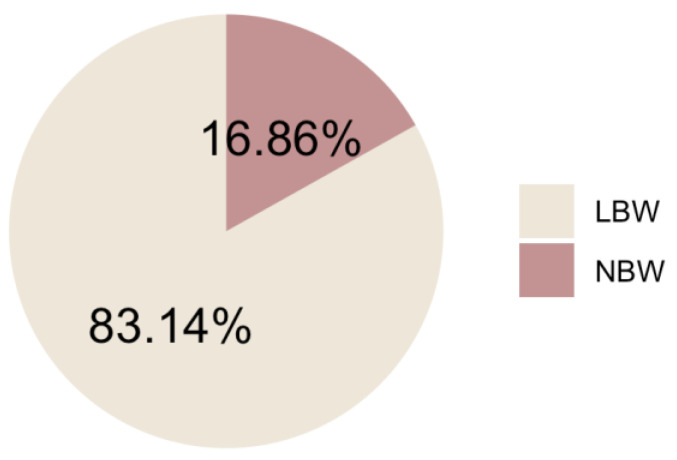
Percentage of LBW babies in India in 2019–2021.

**Figure 2 children-10-01271-f002:**
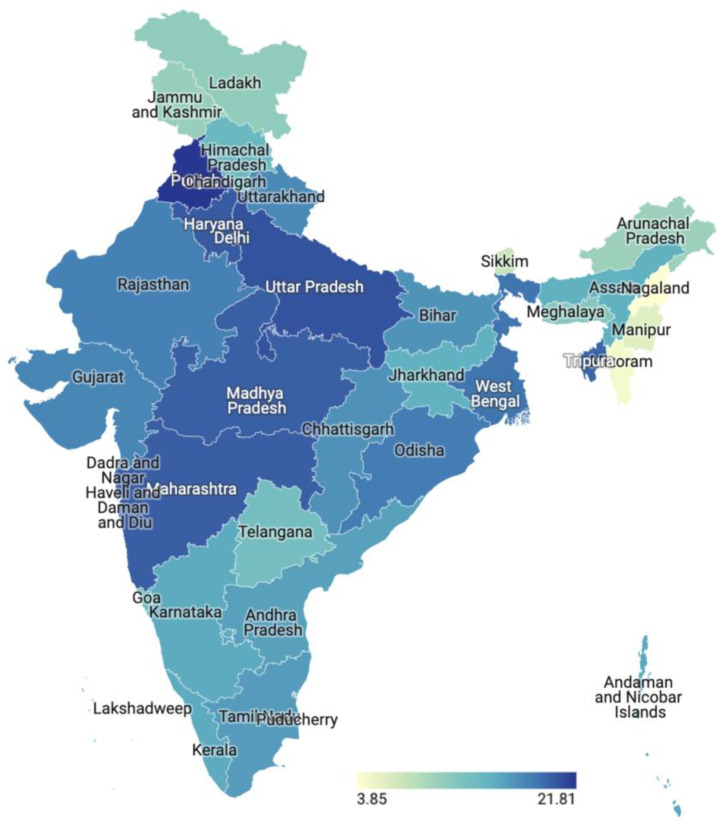
State-level differences in low birthweight.

**Figure 3 children-10-01271-f003:**
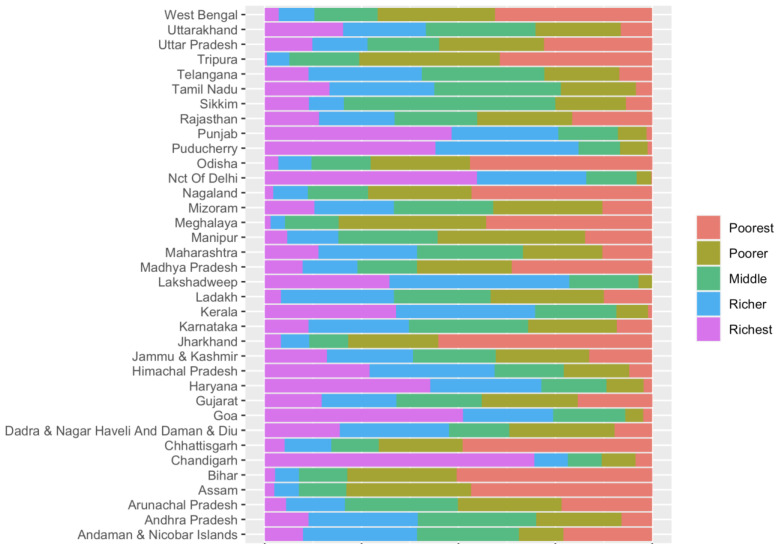
Distribution of low birthweight based on household wealth status in the states and unions.

**Figure 4 children-10-01271-f004:**
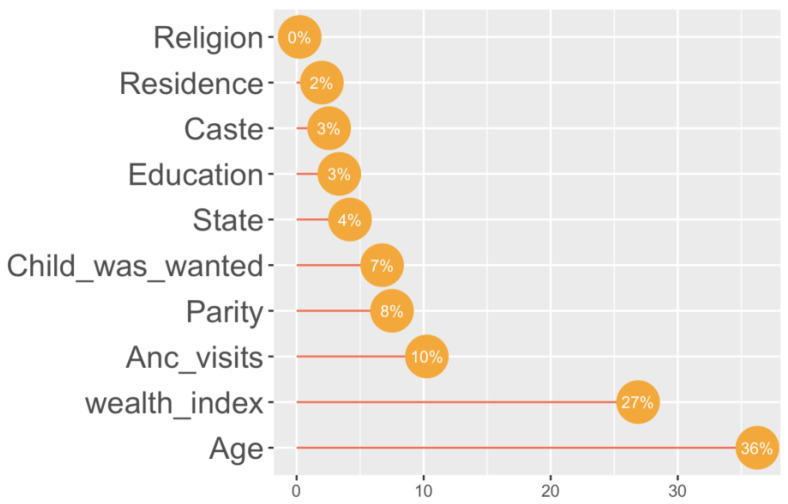
Variable importance plot.

**Table 1 children-10-01271-t001:** Sample characteristics (N = 161,596).

	Total	NBW	LBW	*p*-Value
Type of place of residence: urban	22.3 (22.1; 22.5)	22.5 (22.3; 22.8)	21.2 (20.7; 21.7)	
Type of place of residence: rural	77.7 (77.5; 77.9)	77.5 (77.2; 77.7)	78.8 (78.3; 79.3)	<0.001
Wealth index combined: poorest	23.0 (22.8; 23.2)	22.3 (22.1; 22.5)	26.8 (26.3; 27.3)	
Wealth index combined: poorer	22.7 (22.5; 22.9)	22.5 (22.3; 22.7)	23.6 (23.0; 24.1)	
Wealth index combined: middle	20.2 (20.0; 20.4)	20.4 (20.2; 20.7)	19.1 (18.6; 19.6)	
Wealth index combined: richer	18.5 (18.3; 18.7)	18.8 (18.6; 19.0)	17.0 (16.6; 17.4)	
Wealth index combined: richest	15.6 (15.4; 15.8)	16.0 (15.8; 16.2)	13.6 (13.2; 14.0)	<0.001
ANC visits: inadequate	38.5 (38.3; 38.7)	37.9 (37.6; 38.1)	41.7 (41.1; 42.3)	
ANC visits: adequate	61.5 (61.3; 61.7)	62.1 (61.9; 62.4)	58.3 (57.7; 58.9)	<0.001
Wanted one last child: wanted one then	92.9 (92.8; 93.0)	93.1 (93.0; 93.3)	91.9 (91.5; 92.2)	
Wanted one last child: wanted one later	3.7 (3.6; 3.8)	3.6 (3.5; 3.7)	4.3 (4.0; 4.5)	
Wanted one last child: did not want more children	3.4 (3.3; 3.4)	3.2 (3.2; 3.3)	3.9 (3.6; 4.1)	<0.001
Respondent’s current age, mean (SD)	27.40 (5.13)	27.49 (5.12)	26.98 (5.16)	<0.001
Highest year of education, mean (SD)	4.25 (1.67)	4.26 (1.67)	4.20 (1.66)	<0.001
Total number of children ever born, mean (SD)	2.16 (1.28)	2.17 (1.28)	2.15 (1.30)	0.02
Caste: scheduled caste	21.0 (20.8; 21.2)	20.5 (20.2; 20.7)	23.9 (23.4; 24.5)	
Caste: scheduled tribe	20.2 (20.0; 20.4)	20.7 (20.5; 21.0)	17.5 (17.1; 18.0)	
Caste: OBC	40.7 (40.5; 41.0)	40.7 (40.4; 41.0)	40.9 (40.3; 41.5)	
Caste: none	18.0 (17.8; 18.2)	18.1 (17.9; 18.3)	17.6 (17.1; 18.1)	<0.001
Religion: Hindu	74.7 (74.5; 74.9)	73.9 (73.6; 74.1)	78.7 (78.2; 79.2)	
Religion: Muslim	14.0 (13.8; 14.2)	14.2 (14.0; 14.3)	13.2 (12.8; 13.6)	
Religion: Christian	7.0 (6.9; 7.1)	7.6 (7.5; 7.8)	3.9 (3.7; 4.2)	
Religion: other	4.3 (4.2; 4.4)	4.4 (4.2; 4.5)	4.2 (3.9; 4.4)	<0.001

**Table 2 children-10-01271-t002:** Risk factors of low birthweight in India, 2019–2021.

	Overall	Urban	Rural
Residency (urban)	ref	-	-
Rural	0.93 *** [0.90,0.97]		
Wealth quintile (poorest)	ref	ref	ref
Poorer	0.90 *** [0.86,0.93]	0.97 [0.82,1.15]	0.89 *** [0.86,0.93]
Middle	0.81 *** [0.78,0.84]	0.86 [0.73,1.00]	0.81 *** [0.77,0.84]
Richer	0.79 *** [0.76,0.82]	0.85 * [0.73,0.99]	0.78 *** [0.74,0.82]
Richest	0.74 *** [0.70,0.78]	0.77 *** [0.66,0.90]	0.75 *** [0.70,0.79]
ANC visits (inadequate)	ref	ref	ref
Adequate	0.92 *** [0.90,0.95]	0.92 ** [0.87,0.97]	0.92 *** [0.90,0.95]
Wanted last child (wanted it then)	ref	ref	ref
Wanted later	1.11 *** [1.05,1.18]	1.20 ** [1.06,1.35]	1.08 * [1.01,1.16]
Did not want more children	1.15 *** [1.07,1.24]	1.21 ** [1.05,1.39]	1.13 ** [1.03,1.22]
Age at birth	0.99 *** [0.98,0.99]	0.99 ** [0.99,1.00]	0.98 *** [0.98,0.99]
Years of education	0.99 *** [0.98,0.99]	0.98 * [0.97,1.00]	0.99 ** [0.98,1.00]
Total number of children ever born	0.98 ** [0.96,0.99]	1.02 [0.99,1.05]	0.97 *** [0.95,0.99]
Caste (scheduled caste)	ref	ref	ref
Scheduled tribe	0.81 *** [0.78,0.85]	0.79 *** [0.70,0.90]	0.81 *** [0.77,0.85]
OBC	0.91 *** [0.88,0.94]	0.96 [0.89,1.03]	0.89 *** [0.86,0.92]
None	0.93 *** [0.89,0.96]	0.91 * [0.84,0.98]	0.94 ** [0.89,0.98]
Religion (Hindu)	ref	ref	ref
Muslim	0.92 *** [0.88,0.96]	0.90 ** [0.83,0.97]	0.93 ** [0.88,0.98]
Christian	0.57 *** [0.53,0.62]	0.54 *** [0.45,0.63]	0.59 *** [0.54,0.64]
Other	0.98 [0.92,1.04]	0.89 [0.77,1.03]	1.00 [0.93,1.08]

N.B. Cells represent risk ratios with 95% confidence intervals. Level of significance: * *p* < 0.05, ** *p* < 0.01, *** *p* < 0.001.

## Data Availability

Data are available from the DHS program website.

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
