# Peer review of "Household Wealth Gradient in Low Birthweight in India: A Cross-Sectional Analysis"

_children, 2023, doi:10.3390/children10071271_

Round 1
Reviewer 1 Report
Dear authors,
I have now completed the review of the manuscript titled "Household wealth gradient in low-birthweight in India: a cross-sectional analysis."
In the present study, the authors analyzed the differences in the burden of low-birthweight by household wealth status in India using data from the latest National Family Health Survey (NFHS 2019-21).
The manuscript is interesting and, in general, fair written. In my opinion, the disparities in birth weight by household wealth status seen in this analysis seem to be a reflection of the well-documented link between socioeconomic status and health outcomes.
However, I still have some suggestions to further improve the quality of the manuscript.
1. The introduction section introduced some relevant articles. Please explain the results or summarize with effect sizes.
2. I suggest authors clarify how other researchers can obtain the original data with data availability statements.
3. Description of difference in access to helathcare can be improved. Wealthier families often have better access to quality prenatal care, which can significantly affect birth outcomes. Routine prenatal care can help manage health conditions and detect potential complications before they become severe. Conversely, lower-income families may have limited access to these services due to financial constraints or lack of healthcare facilities in their areas.
4. In the ‘Analysis’ section, I suggest authors to refer to some articles on why authors selections are proper statistical methods. For example, methods for testing statistical differences between groups; regression analysis for continuous independent variables; etc...
5. What is the future scope of the proposed research, authors have described the limitations in a good way, and I suggest that these can be the future scope of the work.
Author Response
- The introduction section introduced some relevant articles. Please explain the results or summarize with effect sizes.
Response: Thanks for this suggestion. I have added some more studies in the introduction.
- I suggest authors clarify how other researchers can obtain the original data with data availability statements.
Response: This was mentioned in the introduction. The data were obtained from the National Family Health Survey which is accessible to all registered users through the DHS program website.[21]
- Description of difference in access to helathcare can be improved. Wealthier families often have better access to quality prenatal care, which can significantly affect birth outcomes. Routine prenatal care can help manage health conditions and detect potential complications before they become severe. Conversely, lower-income families may have limited access to these services due to financial constraints or lack of healthcare facilities in their areas.
Response: Thank you for your valuable input. I fully agree that wealthier families in India often have better access to quality prenatal care, which can significantly affect birth outcomes. Limited access to healthcare services due to financial constraints or lack of healthcare facilities is a major issue for lower-income families in India. In this study, I have attempted to account for this by including household wealth as a variable in the analysis and added more description in this in the discussion section.
- In the ‘Analysis’ section, I suggest authors to refer to some articles on why authors selections are proper statistical methods. For example, methods for testing statistical differences between groups; regression analysis for continuous independent variables; etc…
Response: Thank you for your suggestion. I described the statistical methods in detail, but I agree that referencing relevant literature on statistical methods would improve the clarity. I have cited papers to ensure that our methods are appropriate and properly explained.
- What is the future scope of the proposed research, authors have described the limitations in a good way, and I suggest that these can be the future scope of the work.
Response: Thank you for your comment. I agree that our study has limitations, and I hope that these findings will inspire further research into the determinants of low birthweight in India. I believe that future research could explore more in-depth factors associated with low birthweight, including maternal nutrition, exposure to pollution, and access to healthcare services.
Reviewer 2 Report
Dear Author,
This is an interesting manuscript about how the social status and family income can influence the birth weight of children and subsequently their health, as well as how the burden of a low birth weight of children will put a burden on society. Although the topic of the study is an important one, the current manuscript must be redone because there are many ambiguities regarding the Methodology. In any case, I have a few remarks to make.
Abstract - line 15-16 - the CI ar not necessary here.
Both in the Abstract and in the rest of the manuscript, the second person plural - "we" is used as an expression. I see only one author of the manuscript!
Methods - the survey was conducted by the author of this manuscript? I don't understand why the cited survey (18) is described in the Methods chapter?!
-Line 87 - "2.5 kilograms" is not necessary
-Line 85- What study is this about? Is this manuscript an original article or a literature review?
-Figure 2 - is this figure original? If not, please add a reference and the permission to use it. The same for Figure 3.
The Discussion section in well written.
Author Response
Abstract - line 15-16 - the CI ar not necessary here.
Response: This was corrected. Thank you.
Both in the Abstract and in the rest of the manuscript, the second person plural - "we" is used as an expression. I see only one author of the manuscript!
Response: This was corrected. Thank you.
Methods - the survey was conducted by the author of this manuscript? I don't understand why the cited survey (18) is described in the Methods chapter?!
Response: Thanks for this comment. Some journals require sampling description of the surveys even for secondary datasets. We are open to making changes in this section as per the journal guidelines.
-Line 87 - "2.5 kilograms" is not necessary
Response: The word document doesn’t show any line number, b
-Line 85- What study is this about? Is this manuscript an original article or a literature review?
Response: This was a first-hand analysis of data from the DHS program. It is not a review study.
-Figure 2 - is this figure original? If not, please add a reference and the permission to use it. The same for Figure 3.
Response: Yes, all the figures are original.
The Discussion section in well written.
Response: Thank you.
Round 2
Reviewer 1 Report
All comments have been addressed. Thank you to the authors and editors for considering my opinion on this manuscript.
Author Response
Thank you for the time to review the article!
Reviewer 2 Report
Dear Author,
Although you stated that you corrected problem no. 1 (CI is not necessary in the abstract), in reality you left it the same. The same thing at point no. 4 (line 87, now line 111 - 2.5 kg is not necessary). Regarding the rest of the manuscript, some improvements can be seen.
Author Response
My apologies for the oversight. I uploaded the wrong file, which is now corrected. Thanks indeed for you time to review the article!